# Two-Sex Life Table Analysis for Optimizing *Beauveria bassiana* Application against *Spodoptera exigua* (Hübner) (Lepidoptera: Noctuidae)

**DOI:** 10.3390/jof10070469

**Published:** 2024-07-04

**Authors:** Junaid Zafar, Rana Fartab Shoukat, Zhanpeng Zhu, Dongran Fu, Xiaoxia Xu, Fengliang Jin

**Affiliations:** State Key Laboratory of Green Pesticide, “Belt and Road” Technology Industry and Innovation Institute for Green and Biological Control of Agricultural Pests, College of Plant Protection, South China Agricultural University, Guangzhou 510642, China; jz_jaam@yahoo.com (J.Z.); ranafartab@gmail.com (R.F.S.); zhuzhanpeng@stu.scau.edu.cn (Z.Z.); dealanfu@stu.scau.edu.cn (D.F.)

**Keywords:** *Spodoptera exigua*, beet armyworm, biocontrol, life table, insect-pest, *Beauveria bassiana*

## Abstract

*Spodoptera exigua* (Hübner) (Lepidoptera: Noctuidae) is a highly dispersive, polyphagous insect pest that severely defoliates crops. Excessive reliance on synthetic insecticides leads to ecological pollution and resistance development, urging scientists to probe eco-friendly biopesticides. Here, we explore the virulence of an entomopathogenic fungus, *Beauveria bassiana*, against *S. exigua*, resulting in 88% larval mortality. Using an age–stage, two-sex life table, we evaluated the lethal and sublethal effects of *B. bassiana* on the demographic parameters of *S. exigua*, including survival, development, and reproduction. Sublethal (LC_20_) and lethal concentrations (LC_50_) of *B. bassiana* impacted the parental generation (F_0_), with these effects further influencing the demographic parameters of the first filial generation (F_1_). The infected F_1_ offsprings showed a reduced intrinsic rate of increase (*r*), mean generation time (*T*), and net reproduction rate (*R*_0_). Larval developmental duration varied significantly between the control (10.98 d) and treated groups (LC_20_: 10.42; LC_50_: 9.37 d). Adults in the treated groups had significantly reduced lifespans (M: 8.22; F: 7.32 d) than the control (M: 10.00; F: 8.22 d). Reduced fecundity was observed in the *B. bassiana*-infected groups (LC_20_: 313.45; LC_50_: 223.92 eggs/female) compared to the control (359.55 eggs/female). A biochemical assay revealed elevated levels of detoxification enzymes (esterases, glutathione S-transferases, and acetylcholinesterase) in the F_0_ generation after *B. bassiana* infection. However, the enzymatic activity remained non-significant in the F_1_ generation likely due to the lack of direct fungal exposure. Our findings highlight the enduring effects of *B. bassiana* on the biological parameters and population dynamics of *S. exigua*, stressing its use in eco-friendly management programs.

## 1. Introduction

Life tables are powerful tools for studying arthropod population dynamics [1]. Traditional age-specific life tables that solely consider females [2] can misrepresent the life history traits [3]. However, recent advancements in age- and stage-specific life table methodologies highlight the importance of incorporating both age–stage and two-sex-based data. This integrated approach provides a more accurate representation of life history parameters for organisms with complex developmental stages, such as insects [4,5]. Age–stage, two-sex life table analysis employs a comprehensive (often daily) schedule to track survival rates for both male and female individuals across a life cycle. This method facilitates the estimation of crucial life history characteristics, including age-specific survival, fecundity, and generation time [6]. Notably, these parameters can be assessed in response to various environmental factors, such as constant or fluctuating temperatures, pesticide resistance, and insect–pathogen interactions for biological control [7,8,9].

The beet armyworm, *Spodoptera exigua* (Hübner) (Lepidoptera: Noctuidae), a widely distributed polyphagous agricultural insect pest, causes severe defoliation of fiber, food, and flower crops, leading to substantial economic damage [10,11]. While synthetic insecticides remain the primary methods of pest control, their imprudent use can trigger insecticide resistance, pose health hazards, and disrupt the ecological cycle [12,13]. Therefore, identifying eco-friendly and sustainable control agents remains a priority.

One promising option lies in the use of biocontrol agents, such as entomopathogenic fungi. These fungi encompass diverse genera that are capable of effectively infecting and killing insect pests, exhibiting considerable host specificity that minimizes harm to non-target organisms [14,15]. Furthermore, secondary metabolites produced by entomopathogenic fungi have shown high mortality and antifeedant properties against various insect pests [16,17]. Toxicity tests have shown the effectiveness of several entomopathogenic fungi against the *S. exigua* larvae [18]. The application of *Beauveria bassiana* (6 × 10^7^ conidiospore/mL) efficiently suppressed the larval population of *S. exigua* across two seasons in sugar beet fields [19]. Similar effectiveness of entomopathogenic fungi has been reported in several lepidopteran pests, such as *Spodoptera frugiperda* [20], *Plutella xylostella* [21], and *Helicoverpa armigera* [22].

Insects have an array of specialized detoxification enzymes that participate in the defense against pathogenic toxins [23]. Acetylcholinesterase (AChE) plays a role in terminating neurotransmission by hydrolyzing acetylcholine at synapses, making it a target for many insecticides [24]. Glutathione S-transferases (GSTs) play a crucial role in detoxifying cells and defending against oxidative stress by linking reduced glutathione to electrophilic sites in various natural and synthetic compounds, such as insecticides, allelochemicals, and endogenously activated substances [25]. Esterases (ESTs) play a vital role in insects, catalyzing the breakdown of esters derived from higher fatty acids, which affect flight, as well as degrading metabolic and xenobiotic esters, including insecticides [26]. Changes in enzymatic activities underlie both insecticide resistance and the degradation of secondary toxins produced by entomopathogenic fungi, thereby protecting insects during fungal infection [23,27]. This resistance mechanism stresses the evolutionary adaptations of insects to microbial pathogens, challenging pest control efforts. Understanding the biochemistry of these enzymes is necessary for developing effective pest management strategies.

Continuous monitoring and adaptive strategies are crucial to mitigate resistance and maintain effective pest management practices. An in-depth understanding of the pest demography in relation to entomopathogenic fungi is necessary to formulate effective control strategies. This information leads us toward the identification of optimal stages for the application of biocontrol agents [28,29,30]. Here, we investigate the pathogenicity of three entomopathogenic fungi: *Metarhizium anisopliae*, *B. bassiana*, and *Isaria fumosorosea* against *S. exigua*. Although all larval stages are damaging, we selected the third instar for life table analysis due to its reported fungal susceptibility, ability to continue feeding on foliage, and tendency to disperse toward fruit [19,31]. Furthermore, transgenerational and enzymatic effects were also evaluated to identify the most effective time and stage for applying the chosen entomopathogenic fungus to the control *S. exigua* populations. By understanding demographic parameters and potential resistance, this research aims to update the development of sustainable pest management strategies.

## 2. Materials and Methods

### 2.1. Insect and Pathogenic Fungi

*S. exigua* was obtained from the College of Agriculture, South China Agricultural University, China, and kept in an insecticide-free environment. Larvae were reared on an artificial diet as described previously [32]. Emerging adult insects were allowed to mate in a plastic container provided with a 10% honey solution. Eggs were collected and incubated at 25 ± 2 °C, with a 14:10 light–dark cycle, and 65 ± 5% relative humidity (RH).

Three entomopathogenic fungi (*M. anisopliae*, *B. bassiana*, and *I. fumosorosea*) were grown on potato dextrose agar (PDA) for two weeks and kept in total darkness at 25 ± 1 °C and 75–80% RH. Isolates were passaged multiple times to prevent aging [33]. Conidia were harvested into a 0.05% Tween-80 (Sigma Aldrich P1754, St. Louis, MO, USA) solution, and desired concentrations were prepared.

### 2.2. Bioassays

Five concentrations (3 × 10^4^, 3 × 10^5^, 3 × 10^6^, 3 × 10^7^, and 3 × 10^8^ conidia/mL) for each fungus were tested against 3rd instar larvae, while 0.05% aqueous Tween-80 and distilled water alone were used as controls. To ensure assay effectiveness, germination tests were conducted by following previously described methods. Fungal suspensions were plated onto PDA plates and incubated in complete darkness at 25 °C ± 2 °C for 18 h [34]. Conidium was considered germinated when its germ tube reached a length at least twice the diameter of the conidium itself [35]. Moreover, 40 larvae (10/replicate × 4 replicates/assay) were exposed to each concentration by direct spraying using a fine aerosol sprayer and kept in sterilized containers containing an artificial diet. Mortality data were recorded every 24 h for 7 days and larvae with no movement were considered dead. Fungal pathogenicity was confirmed by placing the dead carcasses in a humid chamber to observe conidial growth. All mortality data were corrected via the Abbott formula. Lethal (LC_50_) and sublethal (LC_20_) concentrations were calculated and experimentally validated. The fungus with the highest mortality and least LC_50_ was selected for subsequent studies.

### 2.3. Effects of B. bassiana on Parental Generation (F_0_)

The 3rd instar larvae (10/replicate × 4 replicates/assay) were exposed to LC_20_ and LC_50_ concentrations of *B. bassiana*, with aqueous Tween-80 serving as the control. Larval mortality, percent pupation, percent emergence, male/female longevity, and fecundity were evaluated.

### 2.4. Transgenerational Effects of B. bassiana on the First Filial Generation (F_1_)

*B. bassiana*-infected (LC_20_ and LC_50_) and control group eggs (*n* = 100) were randomly selected and examined for transgenerational effects. Developmental changes from larvae through pupae to adult emergence were recorded. The emerged adults were counted, paired into opposite sexes, and transferred to cages (1 pair/cage) for data regarding fecundity and survival.

### 2.5. Enzymatic Assays

Detoxification enzyme activity was evaluated following the previously described methods [36]. Briefly, *S. exigua* larvae (*n* = 5) from F_0_ and F_1_ generations were sampled from *B. bassiana* (LC_20_ and LC_50_)-treated and control groups. Samples were homogenized in a 1.5 mL Eppendorf tube with 400 μL of 0.15 M NaCl and subsequently centrifuged. Supernatants were analyzed for AChE, GST, and EST activities [37], with protein concentrations determined using the Bradford method. [38]

#### 2.5.1. AChE

AChE activity was measured by following the previously described methods [39]. Briefly, 15 μL of supernatant was incubated with acetylthiocholine iodide (substrate) in a reaction mixture containing 10 mM DTNB (5,5′-dithiobis-2-nitrobenzoic acid) and 100 mM sodium phosphate buffer (pH = 7.5). Differences in absorbance were measured at a 412 nm wavelength [40].

#### 2.5.2. EST

EST levels were determined by using *p*-nitrophenyl acetate (*p*NPA) and 50 mM phosphate buffer (pH 7.4) as substrate. Absorbance activity was recorded for 4 min at a 405 nm wavelength [37].

#### 2.5.3. GST

GST activity was evaluated by following the previously described methods [41]. Briefly, supernatant from the respective sample, 5 mM glutathione reductase (GR), and 1 mM CDNB (1-chloro-2,4-dinitrobenzene) were mixed with 0.1 M Tris buffer [42]. Absorbance levels were recorded for 4 min at 340 nm wavelengths.

### 2.6. Statistical Analysis

LC_20_ and LC_50_ were calculated using POLO-PC (version 2.0) software [43]. Mortality data underwent one-way ANOVA analysis, with means distinguished by Tukey’s HSD test in Minitab (version 16) software at a 5% significance level. Enzymatic activity was analyzed using ANOVA and means were separated by Tukey’s HSD test. Development, fecundity, and longevity were analyzed using the age–stage, two-sex life table (TWO-SEX MS Chart) [4,5,44]. The bootstrap technique (*n* = 100,000) [45] was used for the mean and standard error of life table parameters [4,5,44].

*R*_0_ (net reproductive rate) is the total offspring produced by an adult throughout its lifetime.
$$R_{0}=\sum_{x=0}^{\infty }l_{x}m_{x}$$*l_x_* indicates the survival likelihood to age *x* for a newly laid egg.
$$l_{x}=\sum_{j=1}^{k}s_{xj}$$*m_x_* is the mean fecundity of individuals at age *x* and can be obtained from the following equation:$$m_{x}=\frac{\sum_{j=1}^{k}s_{xj}f_{xj}}{\sum_{j=1}^{k}s_{xj}}$$

The intrinsic rate of increase (*r*) was assessed using the iterative bisection method and adjusted with the age-indexed Euler–Lotka equation [46]:$$\sum_{x=0}^{\infty }e^{-r\left(x+1\right)}l_{x}m_{x}=1$$

The finite rate was calculated using the following equation:$$\lambda =e^{r}$$

The mean generational time (*T*) indicates the duration for a population to increase *R*_0_-fold at the stable age–stage distribution: $$T=(ln\left.R_{0})\right|r$$*e_xj_* signifies that the expected lifespan of an individual at age (*x*) and stage (*j*) was calculated from the following equation [47]: $$e_{xj}=\sum_{i=x}^{\infty }\sum_{y=j}^{\beta }S_{iy}^{\prime }$$

The age–stage reproductive value (*v_xj_*) (role of individuals of age *x* and stage *j* to the population) was calculated as follows [48,49]:$$V_{xj}=\frac{e^{r\left(x+1\right)}}{s_{xj}}\sum_{i=x}^{\infty }e^{-r\left(i+1\right)}\sum_{y=j}^{k}S_{iy}^{\prime }f_{iy}$$

## 3. Results

### 3.1. Larvicidal Assay and Fungus Selection

Five different concentrations (3 × 10^4^, 3 × 10^5^, 3 × 10^6^, 3 × 10^7^, and 3 × 10^8^ conidia/mL) were tested against *S. exigua*. The highest mortality was recorded at the concentration of 3 × 10^8^ conidia/mL of all three fungi compared to the control. Among these, *B. bassiana* showed the highest larvicidal activity (88%), followed by *M. anisopliae* (69.5%) and *I. fumosorosea* (69%) (*F* = 67, *df* = 6, *p* = 0.002) compared to the control groups (Figure 1). The calculated LC_50_ values are shown in Table 1. *B. bassiana* exhibited the lowest LC_50_ value with the highest larval mortality, and was subsequently selected for life table studies. Furthermore, the LC_50_ (2.5 × 10^3^ conidia/mL) and LC_20_ (3.1 × 10^2^ conidia/mL) values of *B. bassiana* were experimentally validated (Appendix A).

### 3.2. Effects of Lethal (LC_50_) and Sublethal (LC_20_) Concentrations of B. bassiana on F_0_

Concentration-dependent results were observed after treating LC_20_ and LC_50_ concentrations of *B. bassiana* on the F_0_ generation, with larval mortality percentages of 21.92% and 52.33%, respectively. Significant differences in percent pupation were observed in *B. bassiana*-treated groups (47.67% in LC_50_ and 78.01% in LC_20_) and control groups (98.80%). Similarly, the maximum percentage of emergence was recorded in the control group (96.55%) compared to the treated groups (42.4% in LC_50_ and 69.11% in LC_20_). Maximum male longevity of 10.22 days was observed in the control group, which was significantly reduced to 9.71 and 7.11 days in the LC_20_ and LC_50_ treated groups, respectively. Similarly, adult females in the control group had longer lifespans (9.0 ± 0.22 days) compared to those in the treated groups (LC_50_: 6.22 ± 0.23; LC_20_: 8.22 ± 1.54 days) (*p* < 0.05). In addition, reduced fecundity (eggs/female) was observed in the treated females at LC_50_ (273 ± 6.22) and LC_20_ (380.18 ± 7.21) compared to the control (450.11 ± 7.11) (*F*: 84, *df*: 2, *p* < 0.05) (Table 2). Together, these findings illustrate that *B. bassiana* not only causes larval mortality but also disrupts developmental stages, reduces adult longevity, and impairs reproductive capacity in *S. exigua*, thereby influencing both the population dynamics and life history traits.

### 3.3. Effects of Lethal (LC_50_) and Sublethal (LC_20_) Concentrations of B. bassiana on F_1_

#### 3.3.1. Biological Parameters

Transgenerational effects of *B. bassiana* (LC_20_ and LC_50_) on *S. exigua* are presented in Table 3. Eggs from treated females hatched earlier (LC_20_: 1.94 ± 0.24; LC_50_: 1.90 ± 0.30 days) compared to those in the control (2.00 ± 0.00 days). Total larval developmental time (L1–L5) differed significantly between the control (10.98 ± 0.08 days) and treated groups (LC_20_: 10.42 ± 0.04; for LC_50_: 9.37 ± 0.14 days). The pupal duration was also affected by the treatment: 5.98 ± 0.84 days (LC_20_) and 7.27 ± 1.70 days (LC_50_) compared to 6.58 ± 0.31 days in the control group. Male adults in the treated groups had significantly shorter lifespans (LC_20_: 9.65 ± 0.98; LC_50_: 8.22 ± 2.41 days) than controls (10.00 ± 1.35 days). A similar trend was observed in female longevity (control: 8.22 ± 0.43 days; LC_20_: 8.00 ± 0.05; LC_50_: 7.32 ± 1.65 days). Additionally, fungus-treated females laid significantly fewer eggs during their lifespan (LC_20_: 313.45 ± 4.33; LC_50_: 223.92 ± 4.31 eggs/female) compared to the control group (359.55 ± 7.87 eggs/female) (*F*: 97, *p* < 0.001). These findings highlight the transgenerational effects of *B. bassiana* on *S. exigua*, demonstrating accelerated development, altered metamorphosis, shortened adult lifespan, and reduced fecundity.

#### 3.3.2. Population Parameters

Transgenerational effects of *B. bassiana* (LC_20_ and LC_50_) on population parameters of *S. exigua* are presented in Table 4. The control group exhibited a higher intrinsic rate of increase (*r*) (0.2127 ± 0.006) compared to LC_20_ (0.2129 ± 0.003) and LC_50_ (0.2048 ± 0.001). The net reproduction rate (*R*_0_) (offspring/individual) was highest in the control group (143.82 ± 12.11) and decreased in the *B. bassiana*-treated groups (LC_20_: 125.38 ± 7.11; LC_50_: 107.48 ± 6.21) groups. The mean generation time (*T*) was the highest in the control (24.25 days) with substantial reduction (*p* < 0.05) observed in LC_20_ (22.7 days) and LC_50_ (20.99 days)-treated groups. Lastly, a significant difference (per day) was observed in the mean finite rate of increase (*λ*) between the control (1.278 ± 0.02) and LC_50_-treated (1.217 ± 0.05) groups (*F*: 91, *p* < 0.03). These results highlight the transgenerational effects of *B. bassiana* on *S. exigua* population dynamics, revealing not only reduced reproductive capacity and accelerated generation turnover but also diminished intrinsic growth potential.

The age-specific life expectancy (*e_x_*), fecundity (*m_x_*), reproductive value (*v_x_*), and survival rate (*l_x_*) are plotted in Figure 2. The age-specific life expectancy (*e_x_*) approximates the longevity of an individual of age *x*. The results showed that the longevity of *S. exigua* in the LC50-treated individual at age zero (*e*_0_) was lower compared to those in the control group (Figure 2A). The curve of age-specific fecundity (*m_x_*) demonstrated that reproduction began at different ages in different groups (control/treated), with a significantly lower number of eggs recorded in the *B. bassiana*-infected group (Figure 2B). The age-specific reproductive value (*vₓ*) computes the expected future reproductive contribution of an individual at a specific age (*x*) to the population. When the reproduction commenced, consistent reproductive values were observed between the control and *B. bassiana*-treated groups. However, over time, i.e., on the 21st day, the difference became noticeable, with the value of *vₓ* peaking at a maximum in the control group (89.10 days) and remaining lower in the LC_50_ group (50.88 days). This trend continued throughout the remaining days as shown in Figure 2C. At early stages, the survival rate (*l_x_*) curve between the control and treated groups remained non-significant; however, with time, i.e., at 25 days, the survival rate of insects in the LC_50_ group reduced significantly as shown in Figure 2D.

The cohort-specific egg-to-adult survival rates (*s_xj_*) were significantly reduced in *B. bassiana*-infected groups compared to the control group (Figure 3). The *s_xj_* values for adult females and males in the LC_50_ treated group were 0.29 and 0.50, respectively, compared to 0.37 and 0.53 in the control group, indicating a significant effect of *B. bassiana* (Figure 3A–C). The age–stage, two-sex life table also estimated the expected lifespans of different stages in the population (Figure 4). Our analysis of the life expectancy of *S. exigua* across different life stages (*e_xj_*) revealed a higher survival probability of the first larval instar (L1) in the control group than in LC_50_-treated groups. Additionally, the control group exhibited a longer lifespan compared to the treated groups (Figure 4A–C).

The age–stage reproductive value (*v_xj_*) for females was significantly lower in the LC_20_- and LC_50_-treated groups than in the control (Figure 5). Peak reproduction occurred at 13–17 days in the control (Figure 5A), whereas maximum reproduction rates in the LC_20_ and LC_50_ groups were observed at 13–15 and 14–16 days, respectively (Figure 5B,C). In addition, the LC_20_ and LC_50_-treated groups exhibited reduced fecundity, demonstrating the impact of *B. bassiana* on the population dynamics (Figure 6).

### 3.4. Detoxification Enzyme Activity in S. exigua following B. bassiana Infection (F_0_ and F_1_)

AChE, EST, and GST enzymes play crucial roles in the detoxification of xenobiotics in insects, influencing their response to intoxication and potentially contributing to resistance development [23,27]. Figure 7 shows the detoxification enzyme activity in F_0_ and F_1_ generations of *S. exigua* in response to *B. bassiana* (LC_20_ and LC_50_) infection.

#### 3.4.1. AChE

In F_0_ generation, the maximum AChE activity was observed at 24 h in both LC_50_ (13.49 μmol/min/mg protein) and LC_20_ (10.59 μmol/min/mg protein)-treated groups compared to the control (2.4 μmol/min/mg protein), which gradually decreased as time progressed (Figure 7A). F_1_ generation exhibited similar trends (non-significant) in the control and *B. bassiana* (LC_20_ and LC_50_)-treated groups (Figure 7A1).

#### 3.4.2. EST

Maximum EST activity in F_0_ generation was observed at 12 h post-treatment, with the LC_50_ (15.07 μmol/min/mg protein)-treated group exhibiting the highest activity followed by LC_20_ (12.24 μmol/min/mg protein) compared to the control (2.09 μmol/min/mg protein) (Figure 7B). Similar to the AChE enzyme, the EST showed non-significant trends in F_1_ generation (Figure 7B1).

#### 3.4.3. GST

The peak GST activity was observed at 12 h post-treatment in F_0_ generation, with higher activity detected in the LC_50_ (15.5 μmol/min/mg protein) and LC_20_ (8.43 μmol/min/mg protein)-treated groups compared to the untreated control (1.62 μmol/min/mg protein). A noticeable decrease was observed at 36–72 h post-treatment (Figure 7C). However, no significant differences in GST activity were observed in the F_1_ generation (Figure 7C1).

The stable activity of detoxification enzymes in the F_1_ generation could be attributed to the lack of exposure to harmful fungal compounds. However, further studies are required to elucidate the exact mechanisms behind this observation

## 4. Discussion

*B. bassiana* is a potent biocontrol agent, capable of controlling several susceptible and resistant/multi-resistant insect pests [50], including *S. exigua* [19]. Here, we investigate the virulence of *B. bassiana* against *S. exigua* and its impact on fitness parameters across the F_0_ and F_1_ generations using age–stage, two-sex life table analysis. This knowledge can be valuable for developing integrated pest management (IPM) programs by potentially reducing the lethal time and increasing *S. exigua* mortality.

Three entomopathogenic fungi viz. *M. anisopliae*, *B. bassiana*, and *I. fumosorosea*, were evaluated for their effectiveness against the 3rd instar larvae of *S. exigua*. Among them, *M. anisopliae* and *B. bassiana* exhibited high pathogenicity with LC_50_ values recorded at 3.2 × 10^4^ and 2.5 × 10^3^ conidia/mL, respectively. Several studies have examined their potential application as biocontrol agents. Han et al. examined the susceptibility of 2nd instar larvae of *S. exigua* to *M. anisopliae* (FT83) and reported 100% cumulative mortality after treatment with 1 × 10^7^ conidia/mL [51]. Similarly, various isolates of *B. bassiana* proved to be highly effective against *S. exigua* [52]. Studies have explored the virulence of these fungi in other lepidopterans. Kirubakaran et al. reported that virulent strains of *B. bassiana* (MTCC7690) and *M. anisopliae* (MTCC4104) had LC_50_ values of 9.09 × 10^4^ and 6.08 × 10^5^ conidia/mL, respectively, against the rice leaf-folder *Cnaphalocrocis medinalis* (Lepidoptera: Pyralidae) [53]. Similarly, *B. bassiana* exhibited an LC_50_ of 9.3 × 10^5^ conidia/mL against *P. xylostella* (Lepidoptera: Plutellidae) [21] and 1.3 × 10^4^ conidia/mL against *H. armigera* (Lepidoptera: Noctuidae) [54]. The variation in virulence is likely due to the diversification in physiological characteristics of fungal strains, host susceptibility, and experimental conditions [55]. Owing to the lowest LC_50_ and high larvicidal activity, *B. bassiana* was selected for subsequent biochemical and life table analyses.

*B. bassiana* infection disturbed the fitness parameters of *S. exigua*. In the F_0_ generation, concentration-dependent results were observed after being treated with LC_20_ and LC_50_ of *B. bassiana* with reported larval mortality rates of 21.92% and 52.33%, respectively. In addition to lethal effects, decreases in percent emergence, adult longevity, and fecundity were also reported as potential effects of *B. bassiana* infection. Studies have documented similar fitness costs associated with entomopathogenic fungi [56]. Consistent with our findings, sublethal exposure to *B. bassiana* significantly impaired the development and reproductive capacity of various insect pests, including *Nilaparvata lugens* [57], *Eurygaster integriceps* [58], *Sogatella furcifera* [59], and *H. armigera* [60]. These sublethal effects are potentially linked to entomopathogenic fungi-mediated nutritional deficiency [61]. Upon penetrating the insect host, fungal pathogens not only release secondary toxins but also absorb essential sugars from tracheoles, weakening the insects and ultimately impacting their biology and population dynamics [62,63].

The negative effects of *B. bassiana* on *S. exigua* extended beyond the directly exposed generation. In the F_1_ generation, declines in pre-adult duration, adult longevity, and female fecundity were observed, consistent with the findings reported for *B. bassiana* infection in *Cyclocephala lurida* [64] and *M. anisopliae* infection in *H. armigera* [65]. Notably, prolonged pupal duration was observed in the LC_50_ (7.27 days)-treated group compared to the control (6.58); similar to our study, the extended pupal period was reported in *B. bassiana*-exposed *H. armigera* [56,60]. We theorize that the fungal infection imposes physiological and ecological costs on the developing organism, resulting in nutritionally deficient eggs and potentially leading to the continuation of sublethal effects into the F_1_ generation [57,66]. Moreover, the values of the intrinsic rate of increase (*r*), net reproduction rate (*R*_0_), mean length of generation (*T*), and finite rate of increase (*λ*) showed significant reduction in response to lethal and sublethal treatments of *B. bassiana*. Supporting our findings, studies have shown reduced *λ* and *R*_0_ but prolonged *T* values in the *M. anisopliae*-infected tomato leaf miner, *Tuta absoluta* (Lepidoptera: Gelechiidae) [67]. Similarly, lethal and sublethal treatments of *B. bassiana* had detrimental effects on the life-history parameters of *Bactericera cockerelli* in the F_0_ and F_1_ generations [68]. The reduced *r*, *R*_0_, and *λ* values in treatment groups relative to the control indicate the potential impacts of *B. bassiana* infection on the population growth rate and generation. Notably, *r* is considered a particularly sensitive measure of insect response to stressors, as it directly reflects the population’s growth potential [69,70].

Cohort-specific (egg-to-adult) biological parameters (*e_xj_*, *s_xj_*, and *v_xj_*) are vital indicators for assessing the fitness of the insect population. Similar to our findings, the overlapping age–stage survival rate (*s_xj_*) curves between control and treated groups in *M. anisopliae*-infected *Oxycarenus hyalinipennis* were reported [71]. In addition, developmental timings were also affected by fungal infection. Similar trends were also reported in *B. bassiana*-infected *Aedes albopictus* [29], supporting our findings.

AChE, EST, and GST are important physiological metabolic detoxification enzymes that help insects cleanse and resist harmful intoxicants [72]. Our results indicated that the detoxification enzyme activity was significantly elevated in response to *B. bassiana* infection compared to the control. Significant increases in the levels of EST and GST activities in *Dendrolimus tabulaeformis* (Lepidoptera: Lasiocampidae) larvae were closely related to the concentration of conidia and the metabolites of *Beauveria brongniartii* [73], potentially due to the larvae reacting to conidial infection and fungal toxins by relieving oxidative stress. Likewise, in *Diaphorina citri*, higher levels of GST and EST were reported in response to *I. fumosorosea* and *B. bassiana* infections [37], indicating the activation of the antifungal immune response. Exposure to *B. bassiana* and its secondary metabolites has been linked to changes in the activity of AChE in the Sunn pest, *E. integriceps* [74]. A significant increase in AChE activity was observed in the hemolymph of *H. armigera* in response to *B. bassiana* infection [75]. These alterations in AChE activity can be attributed to the production of the secondary metabolite bassianolide by the *Beauveria* spp. Bassianolide, a cyclooligomer depsipeptide, has been detected in the cadavers of silkworm larvae infected with *B. bassiana* [76], demonstrating that its production coincides with infection. Bassianolide can inhibit the acetylcholine receptors of insect muscles, reducing the production of AChE [77]. Furthermore, secondary toxins produced by entomopathogenic fungi are known to induce host cell apoptosis via an increase in oxidative stress and interference with hormonal and mitochondrial signaling while also affecting acetylcholine receptors [78]. However, the enzymatic activity remained non-significant in the F_1_ generation potentially due to the absence or lack of direct fungal exposure. These findings highlight the potential and practical implication of *B. bassiana* for the management of *S. exigua* by targeting the fitness parameters and improving the existing pest control strategies. However, additional research is required to explore the underlying molecular mechanisms.

## 5. Conclusions

In conclusion, our findings demonstrate the promising potential of *B. bassiana* as a pest control agent against *S. exigua*. The fungus caused significant mortality and, through the life table analysis, was shown to disrupt biological parameters and population dynamics, potentially by impairing detoxification mechanisms in the F_0_ generation. Notably, the harmful effects extended beyond the exposed generation, with reduced development time, adult lifespan, and female fecundity observed in the F_1_ generation. These results strongly suggest that *B. bassiana* warrants further investigation as a valuable tool for integrated pest management (IPM) programs.

## Figures and Tables

**Figure 1 jof-10-00469-f001:**
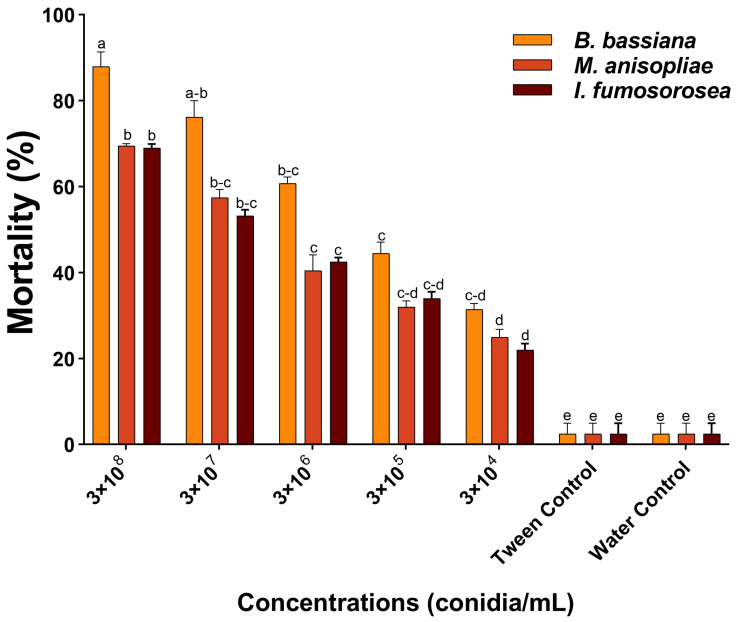
Percentage of larval mortality (3rd instar) of *S. exigua* at different concentrations (3 × 10^4^, 3 × 10^5^, 3 × 10^6^, 3 × 10^7^, and 3 × 10^8^ conidia/mL). While two controls are used. Error bars show 95% confidence intervals (CIs). Different letters indicate significant differences at *p* < 0.05.

**Figure 2 jof-10-00469-f002:**
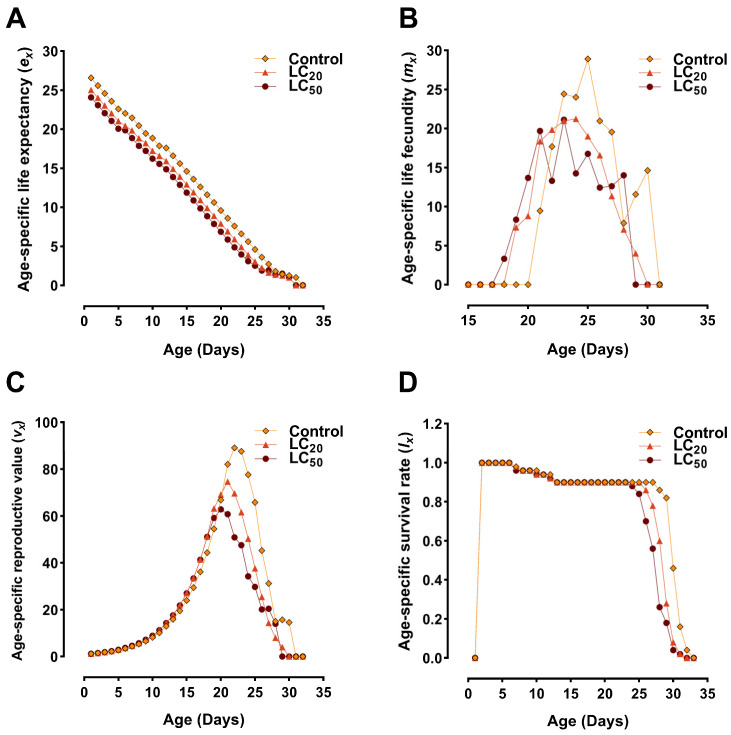
The age-specific (**A**) life expectancy (*e_x_*); (**B**) fecundity *(m_x_*); (**C**) reproductive value (*v_x_*); and (**D**) survival rate (*l_x_*). The sublethal and lethal concentrations of *B. bassiana*-treated groups are represented as LC_20_ and LC_50_, respectively.

**Figure 3 jof-10-00469-f003:**
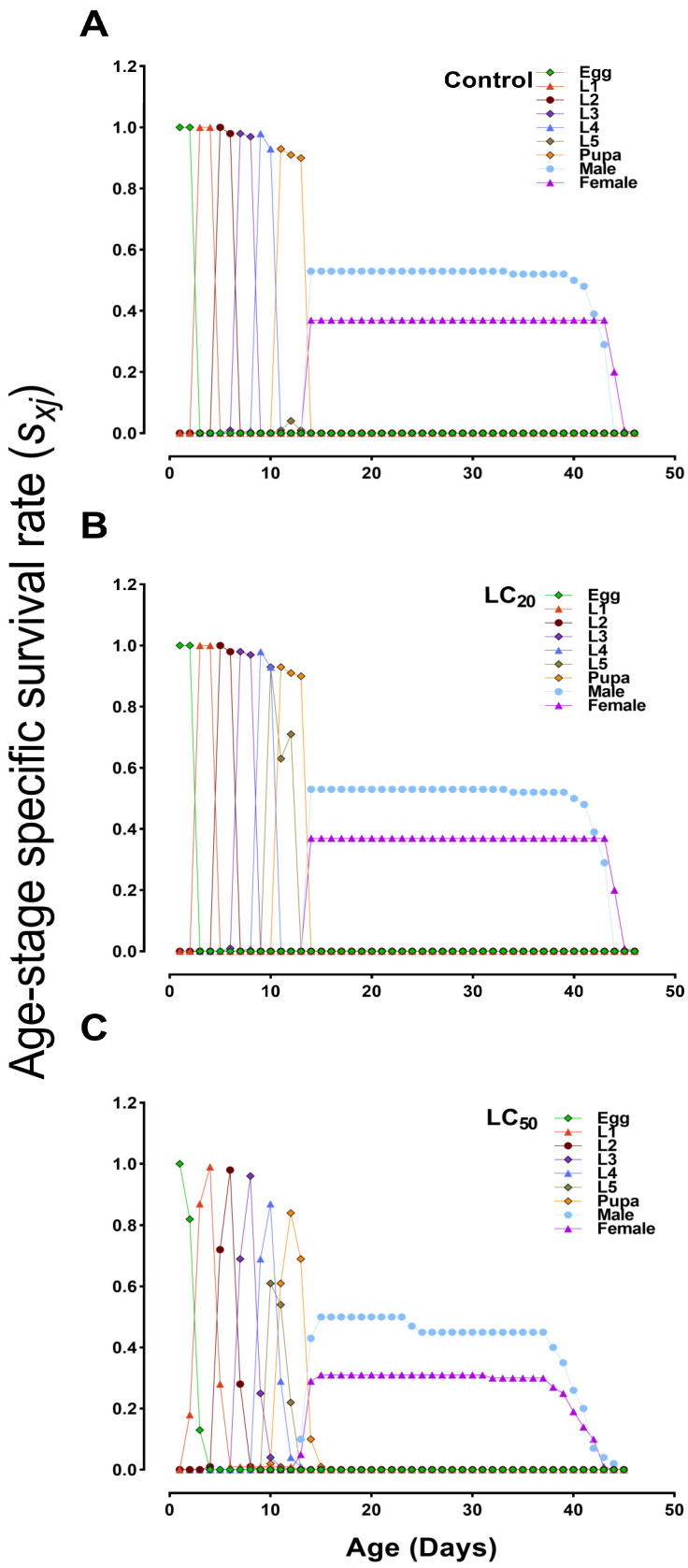
Age–stage-specific survival rate (*s_xj_*) of *S. exigua* after treatment with sublethal (LC_20_) and lethal (LC_50_) concentrations of *B. bassiana*. (**A**) control group; (**B**,**C**) represent LC_20_ and LC_50_-treated groups, respectively. L1–L5 = 1st to 5th instar larvae.

**Figure 4 jof-10-00469-f004:**
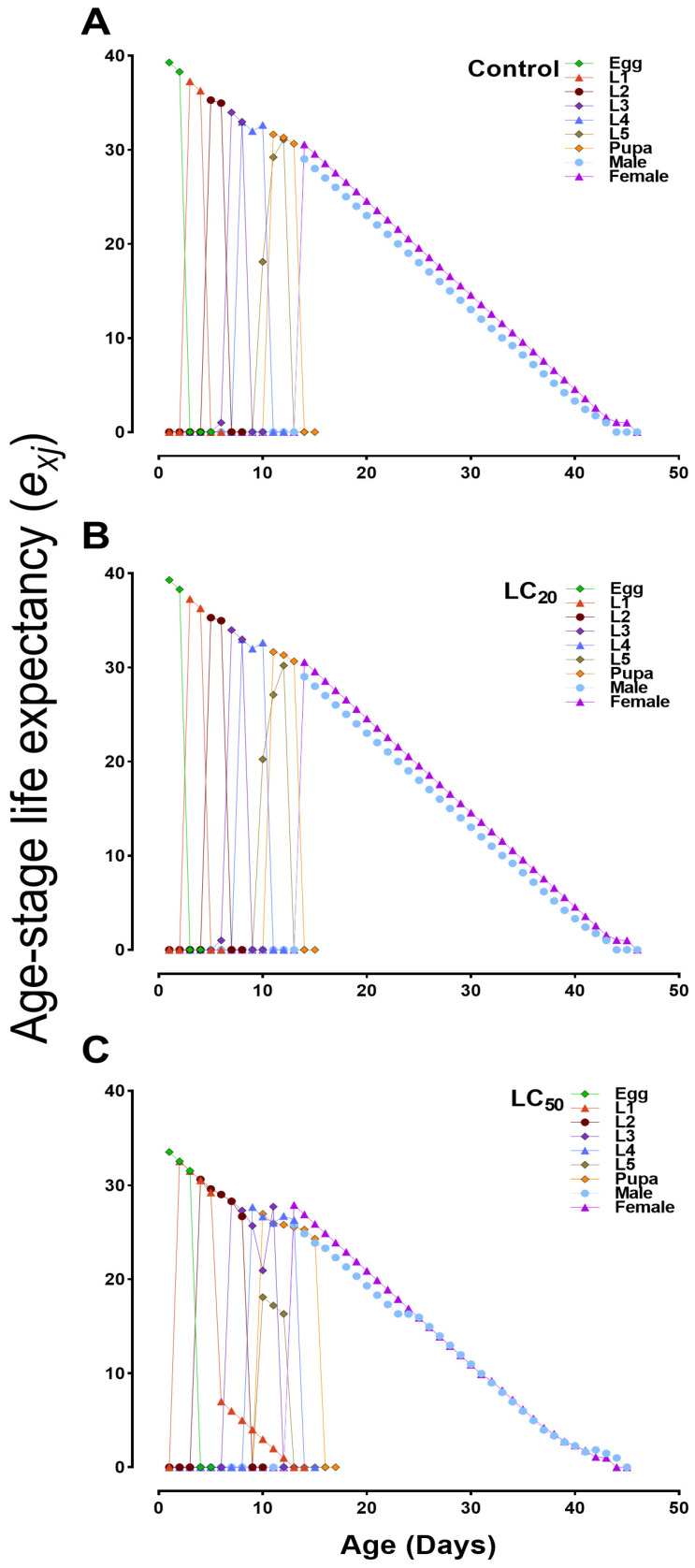
Age–stage life expectancy (*e_xj_*) of *S. exigua* after treatment with sublethal (LC_20_) and lethal (LC_50_) concentrations of *B. bassiana*. (**A**) control group; (**B**,**C**) represent LC_20_ and LC_50_-treated groups respectively. L1–L5 = 1st to 5th instar larvae.

**Figure 5 jof-10-00469-f005:**
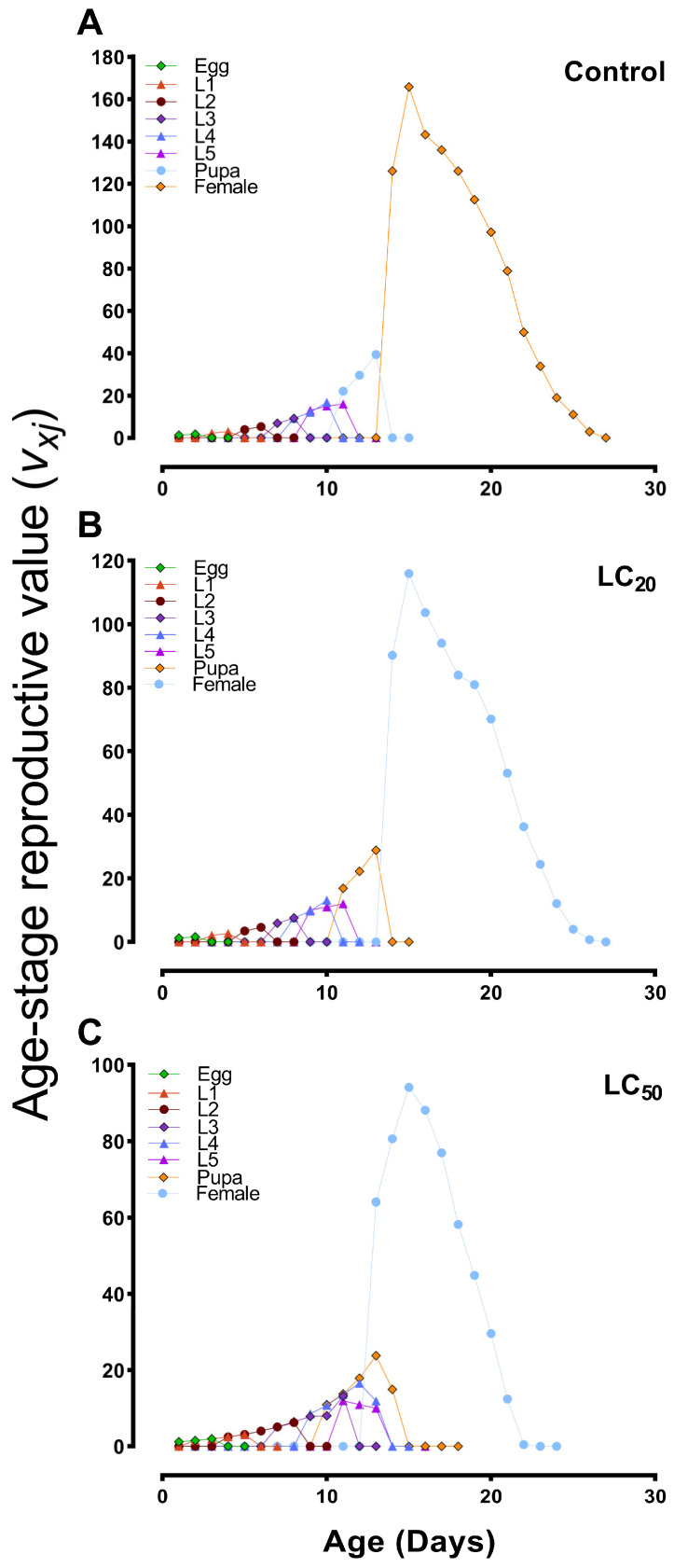
Age–stage reproductive value (*v_xj_*) of *S. exigua* after treatment with sublethal (LC_20_) and lethal (LC_50_) concentrations of *B. bassiana*. (**A**) control group; (**B**,**C**) represent LC_20_ and LC_50_-treated groups. L1–L5 = 1st to 5th instar larvae.

**Figure 6 jof-10-00469-f006:**
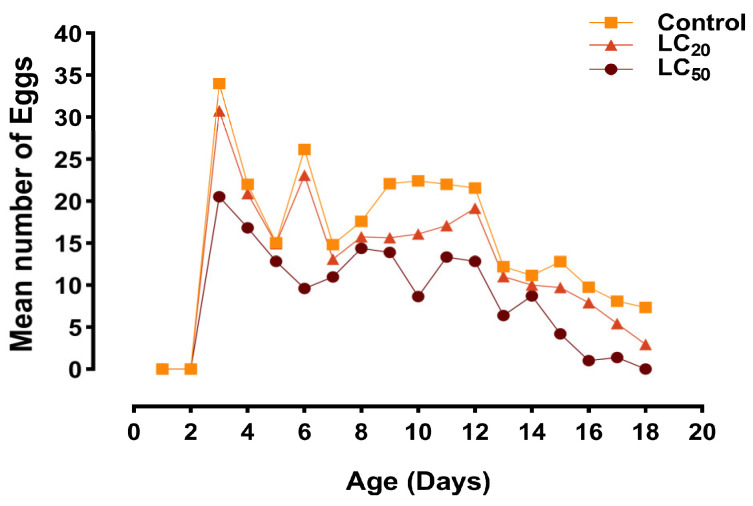
Daily mean number of eggs from *S. exigua* after treatment with *B. bassiana.* The sublethal and lethal concentrations of *B. bassiana* are shown as LC_20_ and LC_50_, respectively.

**Figure 7 jof-10-00469-f007:**
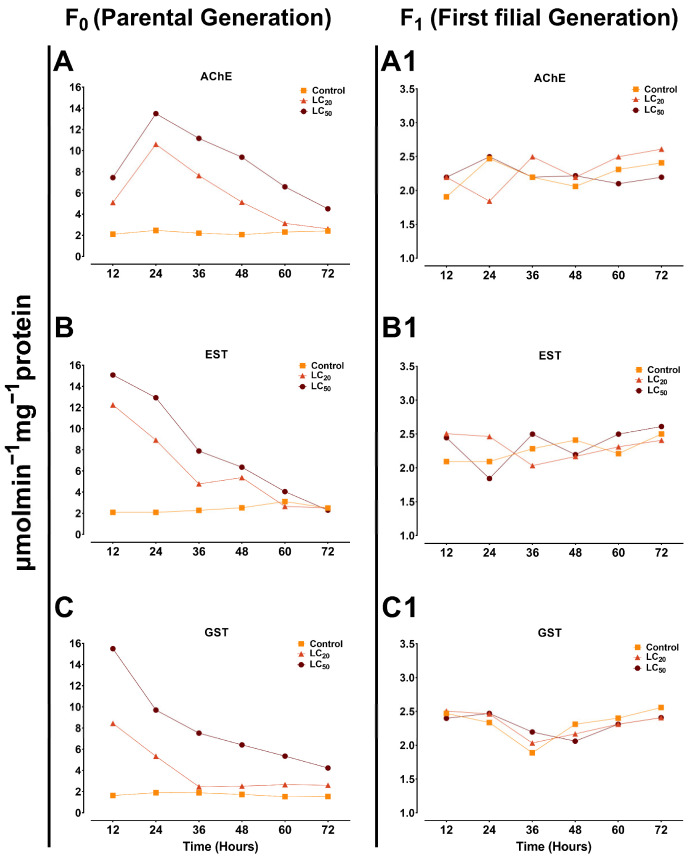
The detoxification enzyme activity in *S. exigua* in response to sublethal (LC_20_) and lethal (LC_50_) concentrations of *B. bassiana*. Figure (**A**–**C**) show the enzymatic activity in the parental/filial (F_0_) generation, while (**A1**–**C1**) show the enzymatic activity in the first filial generation (F_1_). Different colors and symbols are used for different treatments.

**Table 1 jof-10-00469-t001:** Lethal (LC_50_) and sublethal (LC_20_) concentrations of entomopathogenic fungi against *S. exigua* larvae.

Fungi	LC_50_	LC_20_	Slop ± SE	χ^2^	*p*-Value	*df*
*Metarhizium anisopliae*	3.2 × 10^4^	1.0 × 10^2^	0.154 + 0.023	1.87	0.003	5
*Beauveria bassiana*	2.5 × 10^3^	3.1 × 10^2^	0.278 + 0.011	1.25	0.001	5
*Isaria fumosorosea*	2.2 × 10^6^	4.1 × 10^4^	0.357 + 0.054	1.54	0.004	5

Abbreviations: SE = standard error; *df* = degrees of freedom; χ^2^ = chi-square value (confidence Interval 95%).

**Table 2 jof-10-00469-t002:** Influence of *B. bassiana* on parental generation (F_0_) of *S. exigua*.

Parameter	Control	*B. bassiana* (LC_20_)	*B. bassiana* (LC_50_)
	Means ± SE		
Larval mortality	1.2 ± 0.21 ^c^	21.92 ± 1.87 ^b^	52.33 ± 2.11 ^a^
Percent pupation	98.80 ± 2.54 ^a^	78.01 ± 3.15 ^b^	47.67 ± 4.15 ^c^
Percent emergence	96.55 ± 4.11 ^a^	69.11 ± 3.14 ^b^	42.44 ± 2.98 ^c^
Male longevity	10.22 ± 1.32 ^a^	9.71 ± 1.32 ^b^	7.11 ± 2.11 ^c^
Female longevity	9.0 ± 0.22 ^a^	8.22 ± 1.54 ^b^	6.22 ± 0.23 ^c^
Fecundity	450.11 ± 7.11 ^a^	380.18 ± 7.21 ^b^	273 ± 6.22 ^c^

Note: Units are days for male/female longevity. Fecundity (eggs/female). Different letters within rows mark statistically significant differences (*p* < 0.05) between means. Values with the same letter are not statistically significant.

**Table 3 jof-10-00469-t003:** Effects of *B. bassiana* on the first filial generation (F_1_) of *S. exigua*.

Parameters	Control	*B. bassiana* (LC_20_)	*B. bassiana* (LC_50_)
	Means ± SE		
Percent hatching	95.0± 1.01 ^a^	90.22 ± 3.54 ^b^	87.33 ± 2.72 ^c^
Egg duration	2.00 ± 0.00 ^a^	1.94 ± 0.24 ^a^	1.9 ± 0.3 ^b^
L1	2.00 ± 0.01 ^a^	1.84 ± 0.37 ^b^	1.56 ± 0.5 ^c^
L2	1.98 ± 0.01 ^a^	1.92 ± 0.28 ^a^	1.57 ± 0.5 ^b^
L3	2.00 ± 0.07 ^a^	1.9 ± 0.31 ^b^	1.65 ± 0.48 ^c^
L4	2.00 ± 0.05 ^a^	1.94 ± 0.25 ^b^	1.77 ± 0.43 ^c^
L5	3.00 ± 0.02 ^a^	2.82 ± 0.58 ^b^	2.82 ± 0.52 ^b^
Total larval duration	10.98 ± 0.08 ^a^	10.42 ± 0.04 ^b^	9.37 ± 0.14 ^c^
Pupal duration	6.58 ± 0.31 ^b^	5.98 ± 0.84 ^c^	7.27 ± 1.7 ^a^
Adult pre-oviposition period	19.58 ± 1.80 ^a^	18.36 ± 1.13 ^b^	18.49 ± 1.65 ^b^
Female longevity	8.22 ± 0.43 ^a^	8.00 ± 0.05 ^a^	7.32 ± 1.65 ^b^
Male longevity	10.00 ± 1.35 ^a^	9.65 ± 0.98 ^a^	8.22 ± 2.41 ^b^
Fecundity	359.55 ± 7.87 ^a^	313.45 ± 4.33 ^b^	223.92 ± 4.31 ^c^

Note: L1–L5 = 1st to 5th instar larvae. Units are days. Except for fecundity (eggs/female). Different letters within rows mark statistically significant differences (*p* < 0.05) between means. Values with the same letter are not significantly different (*p* > 0.05).

**Table 4 jof-10-00469-t004:** Lethal (LC_50_) and sublethal (LC_20_) effects of *B. bassiana* on the population parameters of *S. exigua*.

Parameters	Control	*B. bassiana* (LC_20_)	*B. bassiana* (LC_50_)
	Means ± SE		
Intrinsic rate of increase (*r*)	0.2127 ± 0.006 ^a^	0.2129 ± 0.003 ^a^	0.2048 ± 0.001 ^b^
Net reproduction rate (*R*_0_)	143.82 ± 12.11 ^a^	125.38 ± 7.11 ^b^	107.48 ± 6.21 ^c^
Mean length of a generation (*T*)	24.25 ± 0.23 ^a^	22.7 ± 0.14 ^b^	20.99 ± 0.04 ^c^
Finite rate of increase (*λ*)	1.278 ± 0.02 ^a^	1.237 ± 0.03 ^b^	1.217 ± 0.05 ^c^
Birth rate (at SASD)	0.2327 ± 0.11 ^b^	0.243 ± 0.02 ^a^	0.243 ± 0.31 ^a^
Survival rate (at SASD)	0.995 ± 0.02 ^a^	0.991 ± 0.07 ^a^	0.993 ± 0.05 ^a^
Death rate (at SASD)	5.35 ± 1.07 ^c^	5.698 ± 1.21 ^b^	6.25 ± 1.57 ^a^

*r* = Intrinsic rate of increase (per days); *R*_0_ = net reproduction rate (offspring/individual); *T* = mean length of a generation (days); *λ* = finite rate of increase (per days); SASD = stable age–stage distribution. Different letters within rows mark statistically significant differences (*p* < 0.05) between means. Values with the same letter are not statistically significant.

## Data Availability

The original contributions presented in the study are included in the article/Appendix A, further inquiries can be directed to the corresponding authors.

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
