# Peer review of "Two-Sex Life Table Analysis for Optimizing Beauveria bassiana Application against Spodoptera exigua (Hübner) (Lepidoptera: Noctuidae)"

_jof, 2024, doi:10.3390/jof10070469_

Round 1

Reviewer 1 Report

The manuscript entitled " Two-Sex Life Table Analysis for Optimizing Beauveria bassiana application against Spodoptera exigua Hübner (Lepidoptera: Noctuidae)” is appropriate for the journal. It is an original and relevant contribution to generate knowledge on the susceptibility of the different stages of development of Spodoptera exigua, is a very complete research work.

Some sections of the manuscript require improvements, for example in the introduction, materials and methods and discussion, which are detailed in the manuscript.

In relation to the virulence parameters, I consider that estimating LC80 is more relevant than LC20

The article can be accepted for publication, attending to the comments of the manuscript.

Some specific comments:

L32- There are previous research works on the evaluation of entomopathogenic fungi against Spodoptera exigua, it is not mentioned in this section, it is necessary to have a complete vision

L90-Indicate exposure method, tt is very important, to test its reproducibility

L164- units

L170- What is the purpose of determining LC20? best CL70, CL80

L206- Units for each parameter

L315- First discuss your results with previous studies on Spodoptera exigua

Reviewer 2 Report

Dear authors,

 The manuscript evaluating the effects of entomopathogenic fungi on Spodoptera exigua is relevant in the context of developing insect management strategies. The introduction is adequate; however, I have made observations to contribute to the improvement of the work. The materials and methods section is robust, with a methodology widely described in the literature. The results are appropriate to the proposed methodology; however, I have also made comments aimed at contributing to the materials and methods and results sections. The discussion is adequate but appears superficial, focusing on comparing the authors' results with those of other authors using unsuitable literature for discussion. In light of the above, here are some observations to help improve the work.

L 57-60: Dear authors, an increase in the level of detoxifying enzymes can occur in insects due to the selection of resistant individuals and other factors such as stress. These enzymes, such as GSTs, ESTs, and CYPs, are strongly linked to the degradation of insecticidal molecules, such as pyrethroids, organophosphates, and neonicotinoids. When addressing the resistance of insects to entomopathogenic fungi, other factors must be considered, such as the mechanical resistance of the integument and peritrophic membrane, phagocytosis, encapsulation, nodulation, and other immunological, biochemical, physiological, and behavioral factors linked to resistance. In this context, I invite the authors to provide a more robust basis for their introduction, justifying why they evaluated the enzymatic action of EST, GST, and AChE, remembering that AChE is more associated with the degradation of acetylcholine (ACh) into choline and acetic acid.

L60-61: There are many studies with Lepidoptera and entomopathogenic fungi. Why do the authors reference manuscripts with Orthoptera and Coleoptera? I recommend looking for more appropriate references.

L66-67: The authors do not mention in the introduction any justification addressing the more susceptible stages of Spodoptera exigua to entomopathogenic fungi. Remember that this insect is holometabolous and although all stages are susceptible to entomopathogenic fungi, the most exposed stage is the larval stage.

L81-84: Dear authors, when reading the abstract, I was led to believe that you would work with formulated products (L15) because you used the term “bioinsecticides.” However, upon reading the materials and methods, I realize that you worked with isolates of Beauveria bassiana, Metarhizium anisopliae, and Isaria fumosorosea. Please adjust the abstract accordingly.

L86-93: Dear authors, was a germination test carried out for these fungi before conducting the bioassay? How were these third instar larvae exposed to concentrations of entomopathogenic fungi? Was the causal agent confirmed by isolating the insects in a sterile humid chamber until the conidia were extruded? How long was mortality checked every 24 hours? Please provide more details in this section.

L94-97: Which instar larvae were exposed? How was the exposure conducted?

L127: Did the authors use the Abbott formula or similar to remove the effect of natural mortality of the larvae and only consider the mortality caused by the different isolates? This information is missing, as removing the effect of natural mortality promotes greater reliability in estimating lethal concentrations.

L158-162: Please provide the F test values, degrees of freedom, and mortality p-value.

L162-164: Provide the confidence intervals between the LC50 and LC20 concentrations.

L175-186: Please provide the F test values, degrees of freedom, and mortality p-value. I recommend that the authors do not merely describe the results in Table 2 to their readers but rather highlight information that is not obvious, presenting the interpretation of these results.

L195-209: Please provide the F test values, degrees of freedom, and mortality p-value. I recommend that the authors do not merely describe the results in Table 3 to their readers but rather highlight information that is not obvious, presenting the interpretation of these results.

L210-224: Please provide the F test values, degrees of freedom, and mortality p-value. I recommend that the authors do not merely describe the results in Table 4 to their readers but rather highlight information that is not obvious, presenting the interpretation of these results.

L277: The authors listed below, and cited in the text, worked with chemical and non-biological insecticides. Search for suitable references to justify the study of AChE, EST, and GST activity.

[42] Fan, R.; Fan, Z.; Sun, Z.; Chen, Y.; Gui, F. Insecticide Susceptibility and Detoxification Enzyme Activity of Frankliniella occidentalis under Three Habitat Conditions. Insects 2023, 14, doi:10.3390/insects14070643.

[43] Chen, X.D.; Seo, M.; Stelinski, L.L. Behavioral and hormetic effects of the butenolide insecticide, flupyradifurone, on Asian citrus psyllid, Diaphorina citri. Crop Protection 2017, 98, 102-107.

L364-373: Please, it would be pertinent for the authors to discuss the role of AChE in the detoxification of enzymes produced by entomopathogenic fungi.

Round 2

Reviewer 2 Report

Dear authors,

I noticed significant changes in the second version of the manuscript. I just ask for one more careful reading of the manuscript to avoid errors like "B. bassianainfection" (L.27).

Congratulations on the work.

just ask for one more careful reading of the manuscript to avoid errors like "B. bassianainfection" (L.27).

Author Response

Dear authors,

I noticed significant changes in the second version of the manuscript. I just ask for one more careful reading of the manuscript to avoid errors like "B. bassianainfection" (L.27).

Congratulations on the work.

Author's Response : Respected reviewer, Thank you for your careful review and for bringing the typographical error in line 27 ("B. bassianainfection") to our attention. We apologize for this oversight and have corrected it to "B. bassiana infection" throughout the manuscript. We've gone through the document again to ensure accuracy and eliminate any remaining errors.

Thank you again for your feedback. We believe it has helped us improve the clarity and quality of our work.

Best Regards,